# Eco-Friendly Capsaicin-Containing Water-Based Antifouling Coatings for Marine Aquaculture

Zeynep Beyazkilic [1],*, Mirko Faccini [2], Ana Maria Escobar [1] and Lorenzo Bautista [1]

[1] Leitat Technological Center, Applied Chemistry & Materials Department, Surface Chemistry Area, 08225 Terrassa, Barcelona, Spain; amescobar@leitat.org (A.M.E.); lbautista@leitat.org (L.B.)
[2] Centro de Excelencia en Nanotecnología (CEN), Leitat Chile, Calle Román Díaz 532, Providencia, Santiago 7500724, Chile; mfaccini@leitat.org
* Correspondence: zbeyazkilic@leitat.org

**Abstract:** Natural antifoulants have received significant interest in the search for non- or less-toxic antifouling coating systems for marine structures. Capsaicin, a natural compound that can be found in chili peppers, is known as an environmentally friendly antifouling agent with an excellent performance and a low environmental impact. However, controlling its release from the coating matrix is still an issue to be solved. With the aim of developing an eco-friendly antifouling system with prolonged long-term activity, in this study, we incorporated capsaicin in combination with dichlofluanid into water-based acrylic coating formulations. The antifouling activity of the resulting coatings was studied by examining the survival behavior of a Gram-negative marine bacterium Aeromonas Salmonoid ATCC 33658, and the release rate of capsaicin from the paint matrix was also assessed. The combination of 0.75 wt.% capsaicin and 0.75 wt.% dichlofluanid into the same matrix showed an antibacterial performance of up to 99.9% cfu reduction with an antibacterial value R 2.5–3 times higher than those obtained with the single biocides at 1.5 wt.%. The association between the two organic biocides created a synergistic effect on the antifouling performance, further resulting in a prolonged activity.

**Keywords:** marine coatings; water-based acrylic coating; capsaicin; dichlofluanid

## 1. Introduction

Marine biofouling is the attachment of marine organisms to the surface of any submerged structure in seawater and is globally one of the major challenges in the marine industry [1]. The growth of these undesired organisms causes negative impacts on economic, environmental, ecological and safety aspects [2]. In the case of marine vessels, biofouling generates surface deterioration and increases frictional drag, which reduces speed, therefore leading to higher fuel consumption and higher emissions of greenhouse gases [3,4]. The settlement of fouling organisms also poses a major problem in marine aquaculture. Biofouling can have many implications in cage nets, such as net deformation and increases in weight. This is a major risk for fish health as it facilitates contact with fouling organisms and reduces oxygen levels and waste flushing [5,6].

To mitigate the impacts of biofouling, a widely used strategy is the application of surface coatings containing biocides, with organotin-based coatings being the most effective in killing fouling organisms [7]. However, such antifoulants were banned in 2008 due to their nontarget biotoxicity and detrimental effect on marine ecosystems [8]. The majority of commercial antifouling coatings currently used in aquaculture contain cuprous oxide as a biocide [5]. Nonetheless, copper is also hazardous and causes adverse impacts on marine creatures such as damage to nerves and gill tissues [9,10]. Accumulated copper sediments in marine farms also negatively affect non-target organisms like invertebrates and algae, producing impairments to embryonic development, movement, and gill health [11]. Additionally, organic booster biocides, which are widely used to strengthen the performance

of copper-based antifoulants, can also impact the marine environment and have received regulatory pressure in various countries [12,13].

Owing to these inevitable drawbacks of metal-based antifoulants and the concerns about complying with the certification standards, there is a great need to develop novel non- or less-toxic antifouling coatings than the currently available copper-based coatings. In particular, salmon farming (Atlantic salmon) has received significant attention regarding biofouling management over recent decades [14,15] and must comply with Aquaculture Stewardship Council (ASC) salmon standards to regulate the use of antifouling coatings [16]. As an alternative to traditional copper-based coatings, ECONEA® (Tralopyril), produced by Jannsen PMP, is one of the few authorized commercial biocides for use in antifouling coatings in the European market. It is defined as an environmentally friendly antifouling agent and can be used with a booster biocide to strengthen the antifouling performance of coatings. On the other hand, research shows that this alternative also has the potential to harm non-target species [17]. It is essential to find a compromise between the concentration of biocidal compounds and their impact on the environment.

Consequently, natural antifouling products as environmentally benign substitutes have been extensively studied. Among these natural compounds, capsaicin (8-methyl-N-vanillyl-6-nonenamide) from terrestrial plants is naturally found in chili peppers and is well known as one of the most promising antifoulants due to its excellent biocidal activity and low environmental toxicity [18,19]. Thus, considerable attention has been given in the literature to investigating the fouling resistance of capsaicin-containing coatings [20–22]. For example, Peng et al. investigated the antifouling activity of capsaicin in solvent-based acrylate paints, comparing it with other biocides such as $Cu_2O$. They showed that $Cu_2O$-free paints with only 0.1% capsaicin as a repellent exhibited a good anti-biofouling activity. However, their long-term performance was severely shortened by the leaching of capsaicin from the paint matrix [20]. Similar results were found by Al-Juhani et al., who studied the antifouling activity of a capsaicin-incorporated solvent-based silicone coating system [23]. To mitigate this problem, other studies have focused on the encapsulation of capsaicin [24] or the synthesis of capsaicin analogues [25,26] as strategies to control the release rate of capsaicin.

In the present study, we were interested in the development of more environmentally friendly antifouling coating alternatives for marine aquaculture, in particular to investigate the prevention of biofouling growth in the marine culture of salmon, because the salmon farming industry has an economic importance in many countries. For example, in Chile, the production of salmon reached 980,000 tons in 2021 (http://www.salmonchile.cl). To this end, we constructed a synergistic antifouling strategy by the combination of two organic biocides, capsaicin and dichlofluanid, to endow the coating formulation with a long-term antifouling performance. Dichlofluanid is known as a common organic booster biocide and is considered an environmentally friendly antifouling agent due to its rapid degradation in seawater [27]. A waterborne acrylic resin was used as a matrix for the preparation of the antifouling coatings due to its water resistance, hardness, and good weathering properties that make it suitable for marine applications [28]. Moreover, the water-based system offers an eco-friendly solution by reducing VOC emissions and being environmentally safe. The antifouling activity of the prepared coatings against the Gram-negative marine bacterium Aeromonas Salmonoid ATCC 33658, which strongly impacts the salmonid population [29], was assessed by the standard ISO 22196:2011 [30] inoculum logarithmic reduction test. The combination of both biocides played a significant role in not only synergistically producing a higher antimicrobial effect, but also decelerating the leakage of capsaicin. To test this hypothesis, we investigated the release rate of capsaicin from the paint matrix in saline water for modelling marine water aquaculture environments.

## 2. Materials and Methods

### 2.1. Materials

Water-based acrylic resin Alberdingk AC 2403 and the additives Disperbyk 2080, BYK 24, BYK 349, and BY7420 ES were kindly provided by BYK-Comindex (Barcelona, Spain). 2-Amino-2-metil-1-propanol (AMP) and Dowanol DPM were purchased from Sigma Aldrich (Madrid, Spain). The pigment $TiO_2$ Kronos 2360 was supplied by Kronos (Barcelona, Spain). Capsaicin ($\geq$95%, from *Capsicum* sp.) and dichlofluanid were purchased from Sigma Aldrich (Madrid, Spain).

AquaNet® Protect was selected as a reference paint because it is an ecofriendly antifouling paint commercialized in Europe. It was supplied by Steen-Hansen and its active substances are ECONEA® (Tralopyril) and zinc pyrithione. All chemicals were used without further purification. Stainless steel 316 was used as a substrate to apply coatings.

### 2.2. Formulation and Preparation of Coatings

The composition of the water-based coating formulations is shown in Table 1. In the first step of the formulation of the biocide free paint, titanium-dioxide-containing pigment slurry was prepared in a mixture of water and dispersant using Dispermat CV3 with a Cowles agitator. Then, the second step consisted of the addition of acrylic resin and the rest of the additives such as the base, antifoam, humectant, rheological additive, and cosolvent into the previously dispersed pigment slurry.

**Table 1.** Composition in wt. % of water-based, biocide-free antifouling coatings.

| Component | Biocide Free | CAP 3 | DIC 3 | CAP 1.5-DIC 1.5 | CAP 1.5 | DIC 1.5 | CAP 0.75-DIC 0.75 |
|---|---|---|---|---|---|---|---|
| Water | 16.07 | 15.59 | 15.59 | 15.35 | 15.79 | 15.79 | 15.79 |
| Disperbyk 2080 | 0.86 | 0.86 | 0.86 | 2.09 | 1.85 | 1.85 | 1.85 |
| AMP | 0.24 | 0.23 | 0.23 | 0.23 | 0.23 | 0.23 | 0.23 |
| BYK 24 | 0.33 | 0.32 | 0.32 | 0.32 | 0.32 | 0.32 | 0.32 |
| Capsaicin | - | 3.00 | _ | 1.50 | 1.50 | _ | 0.75 |
| Dichlofluanid | - | _ | 3.00 | 1.50 | _ | 1.50 | 0.75 |
| Kronos 2360 ($TiO_2$) | 19.69 | 19.10 | 19.10 | 19.10 | 19.17 | 19.17 | 19.17 |
| Alberdingk® AC 2403 | 60.00 | 58.19 | 58.19 | 57.27 | 58.41 | 58.41 | 58.41 |
| BYK 24 | 0.37 | 0.37 | 0.37 | 0.37 | 0.37 | 0.37 | 0.37 |
| BYK 349 | 0.19 | 0.18 | 0.18 | 0.18 | 0.18 | 0.18 | 0.18 |
| Dowanol DPM | 1.66 | 1.61 | 1.61 | 1.58 | 1.62 | 1.62 | 1.62 |
| BYK 7420 ES | 0.57 | 0.55 | 0.55 | 0.54 | 0.55 | 0.55 | 0.55 |
| Total (%) | 100 | 100 | 100 | 100 | 100 | 100 | 100 |

To prepare antifouling paint formulations, the natural antifoulant capsaicin and organic booster biocide dichlofluanid were singularly incorporated and incorporated together in different concentrations. As seen in the table, the antifouling agents (i.e., capsaicin and dichlofluanid) were added into the mixture of water and dispersant (i.e., Disperbyk 2080), followed by the addition of titanium dioxide (Kronos 2360 ($TiO_2$)), and blending with the resin (i.e., Alberdingk® AC 2403 provided by BYK-Comindex, Barcelona, Spain) and additives of the paint system (BYK 24 as defoamer, BYK 349 as wetting agent, BYK 7420 as rheological additive).

The formulations have been applied onto steel substrates using Dr. Blade coatings. Then, the coated substrates were left to dry at room temperature. The ECONEA®-based coating AquaNet® Protect was also applied to the steel substrate for comparison.

### 2.3. Characterization of Coatings

Surface roughness was characterized by a KLA-TENCOR Alpha STEP D 600 3D profilometer. A crosshatch adhesion test was used to evaluate the adherence of the coatings according to ASTM 3359 standard [31]. A lattice pattern with six cuts in the vertical direction was introduced in the coating to the substrate. Pressure-sensitive adhesive tape was applied

to the lattice area. Subsequently, the tape was removed, and the adhesion was classified. A sample classified as 5B is described as having no cut area affected, with the edges of the cut remaining completely smooth. Classifications 4B-1B describe the removal of cut film. An area with less than 5% of film removed is described as 4B; a sample is described as 3B when 15–35% of the film is removed. If the affected area is greater than 65%, it is described as 1B. The surface free energy and contact angles of the coatings were analyzed by a goniometer (Krüss GmbH, DSA100, Barcelona, Spain). The contact angle values were measured using a 2 µL deionized water droplet and a 2 µL diiodomethane droplet. Each water contact angle (WCA) and diiodomethane contact angle (DCA) reported are the average of at least 5 independent measurements. The surface free energy was determined from the contact angles of WCA and DCA. The photos were taken on an iPhone 13, 26 mm $f$1.6 12 MP with 5 cm × 5 cm sized samples.

### 2.4. Antibacterial Activity

The antibacterial activity of the coatings was evaluated against Aeromonas Salmonoid ATCC 33658 via the normative ISO 22196:2011 [30] inoculum logarithmic reduction test. During the test, a thin layer of film, $5 \times 5$ cm$^2$, contained the microorganisms in contact with the sample. Immediately after inoculation, the film was separated from the sample to determine the start time (t = 0). The rest of the samples were incubated for 48 h at 22 °C and analyzed in the same way. This process was performed with control samples and with coatings with or without biocides.

Finally, the number of bacteria was quantified, which allows knowing the logarithmic reduction to assess whether the analyzed samples comply with the regulation standard. The colony-forming unit (cfu) counting method was used. Through the cfu method, only the viable bacteria can be counted and quantified, as this technique excludes dead bacteria.

The value of antibacterial activity (Table 2) was calculated by the following equation:

$$R = (Ut - Uo) - (At - Uo) = Ut - At$$

where $Uo$ is the count of viable bacteria recovered from uncoated samples immediately after inoculation, $Ut$ is the count of the viable bacteria recovered from the uncoated samples 24 h after inoculation, and $At$ is the count of viable bacteria recovered from samples with a coating after 24 h of incubation.

**Table 2.** Efficacy values of antimicrobial activity.

| Efficacy of Antibacterial Property | Antimicrobial Value (R) | % cfu Reduction |
|---|---|---|
| Zero | 0.5 < R | <68% |
| Low | 0.5 < R < 1 | 68% to 90% |
| Medium | 1 < R < 2 | 90% to 99% |
| Significant | 2 < R < 3 | 99% to 99.9% |
| Strong | R > 3 | >99.9% |

### 2.5. Leaching of Biocides

To study the leaching behavior of biocides from the relation between the release amount and the immersion time, $5 \times 5$ cm$^2$ squares of coated substrates were immersed in 150 mL of saline water containing 3 wt% NaCl. The analysis was carried out in a Witeg shaking incubator to stir the samples constantly (120 rpm) at 25 °C. At different time intervals (1 day, 2 d, 4 d, 7 d, 15 d, 21 d, 30 d), aliquots (2 mL) were collected to examine the release of biocides via HPLC analysis with a Waters C18 BEH column (100 mm × 2.1 mm × 2.5 µm). A mixture of acetonitrile (0.1% acetic acid) and DI water (0.2% acetic acid) was used as the mobile phase. The injection volume was 10 µL and the flow was at a constant rate of 400 µL/min. Standard calibration curves were established with the purchased capsaicin and dichlofluanid. Capsaicin and dichlofluanid peaks were detected from the retention time and their amounts were determined from the calibration curve.

The ISO 15181-2 [32] standard was used to test the leaching rate of biocides from antifouling coatings in simulated marine conditions.

The release rate of biocides was calculated from the concentration data obtained by HPLC using the following equation:

$$\text{Release rate } R \left( \mu g \cdot cm^{-2} \cdot day^{-1} \right) \quad \frac{C \times V \times 24}{t \times A}$$

where

$C$ = concentration of capsaicin released into the measuring bath ($\mu g/L$).
$V$ = volume of the water in measuring bath (L).
$t$ = time period of immersion and rotation in the measuring bath (hours).
$A$ = surface area of the test paint ($cm^2$).

The cumulative capsaicin release was calculated using the following equation:

$$R = \sum \frac{(R_i + R_j)}{2}(j - i)$$

where

$R_{x,y}$ = cumulative release of capsaicin from day $x$ to day $y$ ($\mu g \cdot cm^{-2}$)
$R_i$ and $R_j$ = capsaicin release rate between consecutive test days $i$ and $j$ ($\mu g \cdot cm^{-2} \cdot day^{-1}$).
$i$ and $j$ = times from the start of the trial for each pair of consecutive test days (days).

## 3. Results and Discussion

### 3.1. Formulation and Preparation of Coatings

The water-based paints were manufactured using an acrylic resin as a binder and titanium dioxide as a pigment, which are relevant since the acrylic resin has a high resistance in seawater and reduces the solubility of the paint, and titanium dioxide is an effective pigment for a wide range of applications in the paint industry due to its high opacity and covering power [33]. First of all, biocide-free paint was prepared using a white pigment slurry with titanium dioxide. Disperbyk 2080 was chosen as a dispersing agent to stabilize and properly disperse titanium dioxide in an aqueous coating system. Then, the waterborne acrylic dispersion was added to the pigment slurry and the rest of the additives were added to adjust the final properties of the paint formulation. The water-based paint (biocide free) was obtained with suitable dispersion properties. Any significant defects such as cracks or pores were observed on the surface of the biocide-free coating (Figure 1a).

To study the effect of different biocide components and contents, a series of antifouling paint formulations were studied (Table 1). For the first test with the highest biocide contents (CAP 3 and DIC 3), 3.9 wt.% of dispersing agent based on pigment and biocide was used. It was observed that when adding the biocide into the formulation, an increase in dispersing agent was needed to obtain a better dispersion of the pigment and biocide. This was because the quantity, particle size, and chemical structure of biocides introduce an added complexity to the proper dispersion in the water-based paint matrix, even though they are present in small amounts in the total system. Once the dispersion capacity of the biocides was noted, for the final formulations, a 3 wt.% biocide content (CAP 1.5-CAP 1.5) and a 1.5% biocide content (CAP 1.5, DIC 1.5, CAP 0.75-DIC 0.75), with 9.4 and 8.95 wt.% of dispersing agent, were employed, respectively.

The paint formulations were firmly adhered to the steel substrate (Figure 1). Adhesion tests were conducted to measure the adhesive properties of the antifouling coatings. All coatings showed an excellent adhesion property, classified as 5B, where absolutely no film was removed from the surface of the steel substrate (Table 3). The final coating formulations presented as mostly smooth. The surface roughness of the coatings was mostly similar to our reference coating, the commercial paint AquaNet® Protect. As seen from the visual aspect, single-biocide-containing coatings CAP 1.5 and DIC 1.5 exhibited almost homo-

geneous surfaces. The coatings CAP3 and CAP 1.5-DIC 1.5 presented more roughness because the solubility of capsaicin is lower than dichlofluanid in the paint medium.

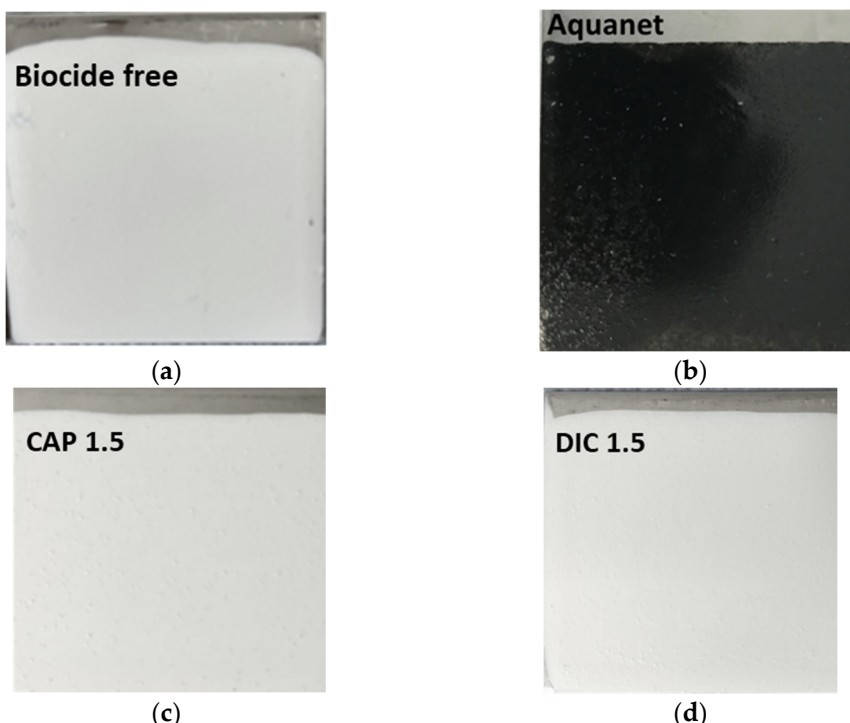

**Figure 1.** (**a**) Biocide-free coating; (**b**) commercial coating AquaNet® Protect; (**c**) CAP 1.5; (**d**) DIC 1.5 (5 × 5 cm²).

**Table 3.** Adhesion test and roughness results of coatings.

| Coating | Adhesion Classification | Roughness |
|---|---|---|
| Biocide Free | 5B | 0.5 ± 0.2 |
| CAP 3 | 5B | 1.7 ± 0.5 |
| DIC 3 | 5B | 1.3 ± 0.6 |
| CAP 1.5-DIC 1.5 | 5B | 1.6 ± 0.5 |
| CAP 1.5 | 5B | 1.2 ± 0.6 |
| DIC 1.5 | 5B | 1.3 ± 0.3 |
| CAP 0.75-DIC 0.75 | 5B | 1.3 ± 0.3 |
| Aquanet | 5B | 1.1 ± 0.4 |

As the surface wettability is an important factor that can affect the antimicrobial activity of an antifouling system, we examined the surface of the coated steel substrates by contact angle analysis (Figure 2). The water contact angles of the biocide-free coating and commercial coating Aquanet were 83° and 93°, respectively, indicating a slightly hydrophobic character. The coatings including biocides showed very small differences in the water contact angle, with values between 75° and 82°, which are close to the water contact angle of the biocide-free analogue. It seems that the incorporation of the biocide into the paint matrix did not significantly affect the surface wettability, which might be due to the small dosage of biocide employed that should be well distributed through the paint matrix. Regarding the behavior in an apolar liquid, diiodomethane, the contact angle values were mostly around or below 60°, indicating a low oleophobic character and a tendency for a moderate attraction for such liquids.

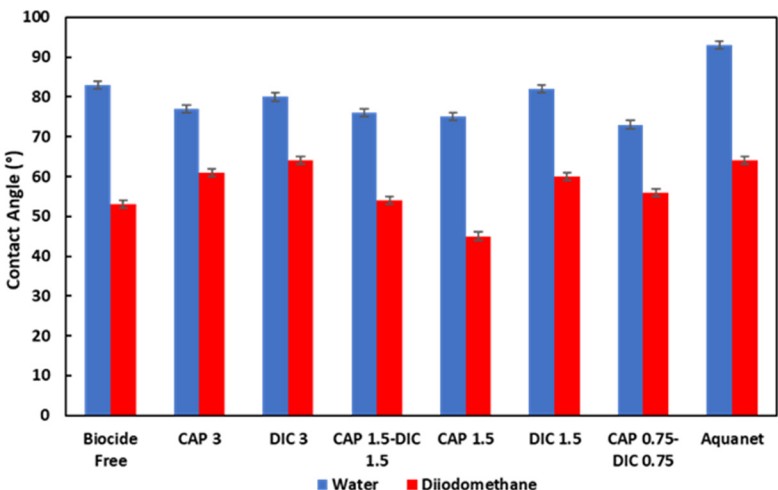

**Figure 2.** Water and diiodomethane contact angles of the coatings.

The surface tension of the coatings developed in this study was found to be between 33 and 40 mN/m (Figure 3), with the water contact angles exceeding 75°, indicating a good water repellence [28] that could have a significant role in the coating quality and aging properties.

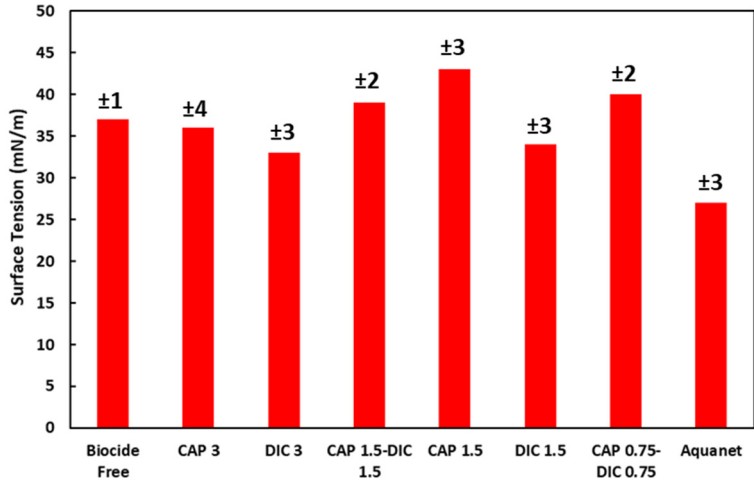

**Figure 3.** Surface tensions of the coatings.

### 3.2. Antibacterial Activity of Coatings

A Gram-negative bacterium, Aeromonas Salmonoid, was used to determine the antimicrobial activity of coatings. It is known as a group of Chilean strains isolated from seawater Atlantic salmon farms. Aeromonas Salmonoid produces systematic infections and affects the health of Atlantic salmon. This has a negative impact on the Chilean economy as the marine culture of Atlantic salmon is of economic importance [29].

Table 4 shows the results of the antibacterial properties of coatings when placed in contact with Aeromonas Salmonoid marine bacteria colonies within 48 h of incubation. It is noticeable that the antibacterial value of the biocide-free control coating is very low, indicating poor antifouling activity, while coating CAP 3 with 3 wt.% capsaicin loading possesses a strong antibacterial performance with a 99.9% cfu reduction.

Although the organic booster biocide dichlofluanid is known particularly to prevent the settlement of algae, diatoms, and other fouling organisms, it also shows a broad spectrum of activity against marine bacteria [34]. When dichlofluanid was incorporated at 3 wt.% loading in our paint system (DIC 3), it resulted in a significant inhibition of bacteria attachment in the range of 99.0–99.9%. Surprisingly, when both biocides were

combined in the same coating matrix at 1.5 wt.% in sample CAP 1.5-DIC 1.5, an outstanding cfu reduction of >99.9% was reached against Aeromonas Salmonid, this being a higher antifouling performance than the DIC 3 coating (incorporating 3 wt.% dichlofluanid).

**Table 4.** Antibacterial performance of coatings.

| Coating | Antimicrobial Value (R) | % cfu Reduction |
| --- | --- | --- |
| Biocide Free | 0.64 | 68–90 |
| CAP 3 | >3.34 | >99.9 |
| DIC 3 | 2.22 | 99.0–99.9 |
| CAP 1.5-DIC 1.5 | >3.34 | >99.9 |
| CAP 1.5 | 1.05 | 90–99 |
| DIC 1.5 | 1.38 | 90–99 |
| CAP 0.75-DIC 0.75 | 3.46 | >99.9 |
| Aquanet | 2.66 | 99.0–99.9 |

To test if the settlement of Aeromonas Salmonid could also be prevented and whether the synergistic effect between capsaicin and dichlofluanid is clearly visible at lower loadings, coating formulations with reduced biocide contents entrapped both individually and in combination were prepared. As shown in Table 4, reducing the content of a single biocide to 1.5 wt.% resulted in coatings CAP 1.5 and DIC 1.5 exhibiting only a medium inhibition, with an antibacterial value R of 1.05 and 1.38, respectively. However, the sample CAP 0.75-DIC 0.75, with a total biocide content of 1.5 wt.% but combining 0.75 wt.% capsaicin and 0.75 wt.% dichlofluanid in the same coating, presents a cfu reduction percentage of 99.9% with an R of 3.46, 2.5–3 times superior to the antibacterial value obtained with the single biocides at 1.5 wt.%. These results suggest the existence of a synergistic effect between capsaicin and dichlofluanid, the combination of which can produce a greater antifouling effect than both compounds used independently at higher concentrations.

Within this context, we also believe that the combination of both biocides could broaden the activity spectrum of the coating against the growth and settlement of not only marine bacteria but also a wide variety of marine organisms, such as diatoms, algae, invertebrates, etc. Additionally, this coating shows more potent inhibition against the growth of Aeromonas Salmonoid than the Econea-based commercial coating Aquanet, used as a reference in this study.

*3.3. Leaching of Biocides*

The release of antimicrobial contents from the paint matrix is a critical factor for the long-term performance of antifouling coatings. To determine the leaching rate of the antifoulants, steel substrates coated with the formulations CAP 1.5 (1.5 wt.% capsaicin content), DIC 1.5 (1.5 wt.% dichlofluanid content), CAP 1.5-DIC 1.5 (1.5 wt.% capsaicin and 1.5 wt.% dichlofluanid in the same coating), and CAP 0.75-DIC 0.75 (0.75 wt.% capsaicin and 0.75 wt.% dichlofluanid in the same coating) were immersed in static cells containing saline water for four weeks. As shown in Figure 4, after this time, the cumulative release of capsaicin of CAP 1.5, CAP 1.5-DIC 1.5, and CAP 0.75-DIC 0.75 reached 67 $\mu g/cm^2$, 50 $\mu g/cm^2$, and 37 $\mu g/cm^2$, respectively. On the other hand, the concentration of dichlofluanid could not be measured in all tested samples. Its concentration probably remained below the quantifiable levels by HPLC because dichlofluanid is prone to degrade within hours in seawater [35].

It is important to note that the cumulative release of capsaicin from paint CAP 1.5, with 1.5 wt.% capsaicin loading, was found to be higher, with a faster release rate compared with the sample CAP 1.5-DIC 1.5 also containing the same amount of capsaicin in addition to 1.5 wt.% dichlofluanid. This result might be due to the physico-chemical interactions between capsaicin and dichlofluanid molecules in the same paint matrix lowering the release rate of capsaicin, thus resulting in a longer-lasting antifouling effect. This observation is also consistent with its strong inhibition capability.

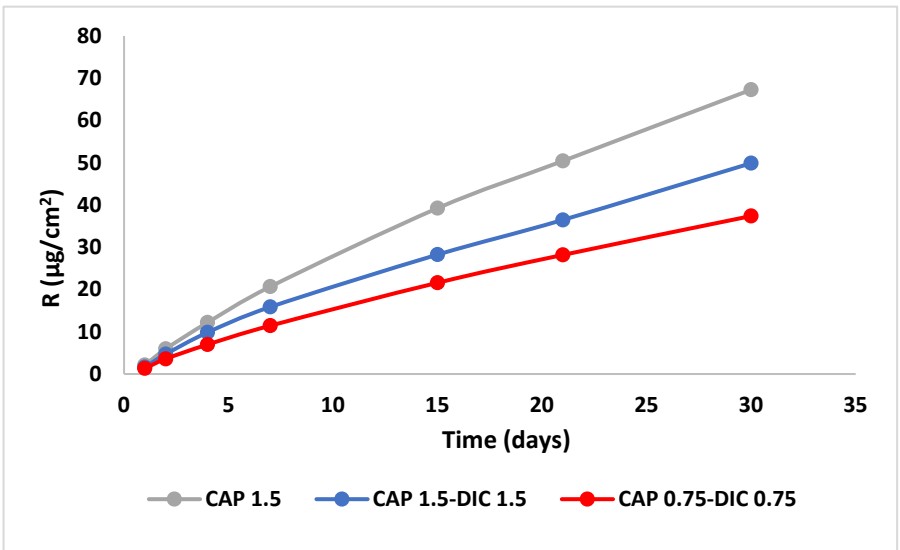

**Figure 4.** Cumulative release of capsaicin per area (R, in $\mu g/cm^2$) from antifouling coatings CAP 1.5, CAP 1.5-DIC 1.5, and CAP 0.75-DIC 0.75.

Capsaicin leaching from CAP 0.75-DIC 0.75 (0.75 wt.% capsaicin + 0.75 wt.% dichlofluanid content), with a bacteria killing efficiency of up to 99.9%, is lower compared to the 1.5 wt.% capsaicin-incorporated coating with a moderate inhibition capacity due to the initial dosage of capsaicin being less (0.75 wt.%).

## 4. Conclusions

In this study, eco-friendly, antifouling, water-based acrylic marine coatings containing capsaicin and dichlofluanid biocides were prepared. The coatings had an excellent adhesion performance on steel surfaces. The antifouling activity of the coating occurs by contact between the coating surface and the marine bacteria that specifically attack salmon. The combination of two organic biocides into the same paint matrix showed a synergistic effect, with an antibacterial performance of up to 99.9% cfu reduction. Moreover, this combination would suggest a broad-spectrum activity on a variety of other marine foulers in addition to bacteria inhibition. It is also worth noting that the combination of capsaicin with dichlofluanid decelerated the leaching of capsaicin out of the coatings and could allow for tuning of the antifouling effect compared to the coating system with capsaicin alone.

**Author Contributions:** Conceptualization, M.F. and L.B.; methodology, Z.B., A.M.E. and L.B.; investigation, Z.B. and A.M.E.; writing—original draft preparation, Z.B.; writing—review and editing, Z.B., M.F., L.B. and A.M.E. All authors have read and agreed to the published version of the manuscript.

**Funding:** Financial funding was provided by the Innovation Fund for Competitiveness of the Chilean Economic Development Agency (CORFO) through the project 13CEI2-21839.

**Institutional Review Board Statement:** Not applicable.

**Informed Consent Statement:** Not applicable.

**Data Availability Statement:** Not applicable.

**Acknowledgments:** The authors acknowledge Advanced Technological Services of Leitat Technological Center for their support in microbiology analyses.

**Conflicts of Interest:** The authors declare no conflict of interest.

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
