# Peer review of "Eco-Friendly Capsaicin-Containing Water-Based Antifouling Coatings for Marine Aquaculture"

_coatings, doi:10.3390/coatings13091616_

Round 1

Reviewer 1 Report

The research presented in publication "Eco-Friendly Capsaicin Containing Water Based Antifouling 2 Coatings for Marine Aquaculture" raises an important topic, however, it requires corrections. My comments below:

Comment 1: “Amino Metil Propanol” Is that the correct name?

Comment 2: 94: “Error! Reference source not found.” Please correct this, there is an error with the reference. This error occurs frequently throughout the work, making it difficult to read.

Comment 3: Table 1: Why does content of Disperbyk 2080 vary significantly between samples?

Comment 4: Is the addition of titanium oxide required for this application?

Comment 5: Figure 1: Add scale and photos for all samples. In the figure's caption, it is also worth adding information whether these photos were taken with a camera.

Comment 6: Results for surface roughness and layer adhesion should be reported. I know you have added a short description of them but they should be shown in graphic form pictures/charts.

Comment 7: Figure 3: There are no standard deviations.

Comment 8: Figure 4 and Table 3 are the same results. Why repeat them?

Comment 9: In my opinion, research should be done on more types of bacteria. If you do not agree with me, it is worth explaining in more detail why research is conducted only for this type and why it is so important.

Comment 10: Figure 5: There are no standard deviations. Please add them. The release is not complete. Why wasn't the research continued?

Comment 11: In my opinion, it is worth adding SEM images to determine whether the drug does not crystallize and does not agglomerate. After dissolution and release, drug holes can increase the corrosion of steel. What do you think? Have you considered adding corrosion results?

Comment 11: It is worth comparing your results with other eco-friendly layers described in the literature.

Author Response

The authors would like to thank the reviewers for making very valuable comments that have improved our research.

Reviewer 2 Report

In this paper, an interesting research was carried out that Capsaicin is used in antifouling coating. The overall scheme design, result analysis and writing of the article are excellent. It is suggested that the author modify the following issues.

1. There are several (Error! Reference 185 source not found) appeared in the manuscript that need to be amended.

2. The title of the article containing ‘Eco Friendly’, but dichlofluanid was used in the experiment and the conclusion also recommends the use of capsaicin with it, which needs to be explained.

The writing meets standards, but there are some unnecessary errors that need to be corrected by the author.

Author Response

(The authors gave the same response as above.)

Round 2

Reviewer 1 Report

The work may be accepted